# LOCAL EXPERT DIFFUSION MODELS FOR EFFICIENT TRAINING IN DENOISING DIFFUSION PROBABILISTIC MODELS

## ABSTRACT

Diffusion models have emerged as a new standard technique in generative AI due to their huge success in various applications. However, their training can be prohibitively time-consuming, posing challenges for small businesses or academic studies. To address this issue, we propose a novel and practical training strategy that significantly reduces the training time, even enhancing generation quality. We observe that diffusion models exhibit different convergence rates and training patterns at different time steps, inspiring our MDM (Multi-expert Diffusion Model). Each expert specializes in a group of time steps with similar training patterns. We can exploit the variations in iteration required for convergence among different local experts to reduce total training time significantly. Our method improves the training efficiency of the diffusion model by (1) reducing the total GPU hours and (2) enabling parallel training of experts without overhead to further reduce the wall-clock time. When applied to three baseline models, our MDM accelerates training $\times 2.7$ - $4.7$ faster than the corresponding baselines while reducing computational resources by 24 - 53%. Furthermore, our method improves FID by 7.7% on average, including all datasets and models.

## 1 INTRODUCTION

Diffusion models have emerged as a powerful new family of generative models for both conditional (Dhariwal & Nichol, 2021; Hertz et al., 2022; Karras et al., 2022; Li et al., 2022; Lugmayr et al., 2022; Nichol et al., 2022; Poole et al., 2023; Rombach et al., 2022; Saharia et al., 2022; Song et al., 2021b) and unconditional (Ho et al., 2020; Nichol & Dhariwal, 2021; Song et al., 2021b) generation tasks, offering notable advantages over existing models, such as generative adversarial networks (GANs (Goodfellow et al., 2014)). These advantages encompass four main aspects (Choi et al., 2022): (1) improved training stability, (2) extensive coverage of data distribution, (3) simple and scalable model structure, and (4) adaptable architecture that facilitates conditional generation (Dhariwal & Nichol, 2021; Ho & Salimans, 2021). The advancements in model design and training strategies (Dhariwal & Nichol, 2021; Dockhorn et al., 2022; Ho et al., 2020; Karras et al., 2022; Nichol & Dhariwal, 2021) have led diffusion models to beat the current state-of-the-art in several fields (Deng et al., 2009; Yu et al., 2015).

However, training large-scale diffusion models is extremely expensive and time-consuming. Training time increases quadratically by the resolution of the dataset. For instance, training a diffusion model on $512 \times 512$ ImageNet (Deng et al., 2009) dataset using a single V100 GPU (Dhariwal & Nichol, 2021) takes up to 1914 days. This substantial training expenses leads to critical delays in deployment within industries and impedes the widespread adoption of diffusion models in small-scale businesses and academia. In this paper, our research objective centers on analyzing the training efficiency of diffusion models.

The training efficiency can be evaluated from two perspectives: (1) the total cost of fully training a model (**TC**), measured in GPU days, and (2) the actual training time (wall-clock time, **WCT**), measured in days. The relationship between TC and WCT can be expressed as TC = WCT $\times$ RT, where **RT** denotes resource throughput, representing the number of distributed GPUs or nodes employed. For example, if a model takes 100 V100 days (TC) to converge, it takes 25 days (WCT)

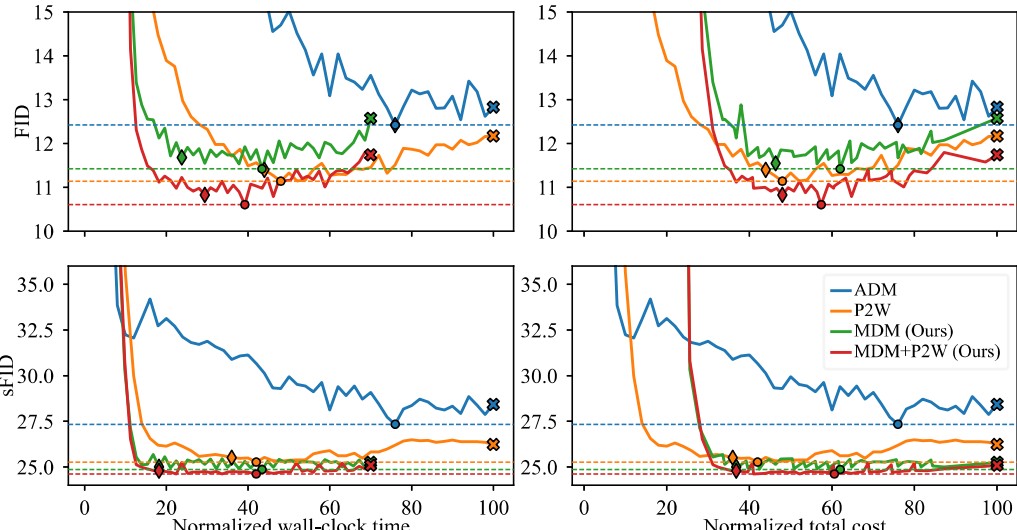

Figure 1: Quantitative evaluation for the normalized WCT (NWCT) and normalized TC (NTC) axes. The total WCT and TC of training the baseline for 500K iterations are set to 100%NWCT and 100%NTC, respectively. The best FID value for each model is denoted by '∘' markers and its value as horizontal dotted lines. The termination point for full iteration is denoted as '×' marker. We determine the model convergence point as the first point where the score difference between adjacent points is smaller than 0.1 for three consecutive sampling points, and at the same time, the score gap to the best FID value is smaller than 0.3 as marked with '◇'.

with four V100 GPUs (RT), assuming ideally distributed training. Considering both TC and WCT are essential when evaluating training efficiency. We aim to devise a method to effectively reduce both TC and WCT by leveraging the intrinsic training patterns of diffusion models.

To reduce WCT, we can increase the RT of the model by parallelizing the training process across multiple modules. However, the increase in RT does not align precisely with the decrease in WCT in practice. This misalignment arises due to computational overhead from communication between devices (Shi et al., 2018; Wu et al., 2022). Dividing the model or batch size also requires an additional algorithm to ensure optimal throughput (Huang et al., 2019; Narayanan et al., 2021) and cannot be done infinitely. This overhead issue is significant, especially when handling large RT. For example, suppose we train a diffusion model with the same batch size. Ideally, if the batch size is split in half between two GPUs, the Wall Clock Time (WCT) should be 50% compared to training with single GPU. However, the actual WCT is around 58% due to the computational overhead. If this situation is extended from inter-GPU to inter-node, this overhead significantly increases.

With this objective in mind, we explore the distinct properties of the training process in diffusion models. We focus on the inherent property of time-independent training in diffusion models. Training each time step $x_t$ is conducted independently (Song et al., 2020) across the entire time step range $t \in (0, T]$ (where $t = T$ represents the fully noisy step). We divide the entire time steps into eight sub-intervals, each assigned to a dedicated expert. Our investigation reveals significant variations in convergence speed among different experts. Notably, the expert handling the fully noisy signal ($t = T$) exhibits the slowest convergence, followed by the expert generating the noise-free signal ($t = 1$). In contrast, experts for middle intervals show faster convergence. We identify that training the entire time steps with a single model results in adverse interactions between different time steps. Ignoring the distinctive nature of diffusion models in their training leads to slow convergence and inferior performance (Sec. 4.1).

Based on this observation, we propose a multi-expert diffusion model (MDM), an algorithm that accelerates training via time step-adaptive local experts. We carefully identify three time intervals, each exhibiting a similar training pattern based on an activation analysis. Then, we train three experts independently, each responsible for each interval. This simple modification to the training strategy enhances the training efficiency of diffusion models. Since MDM consists of multiple independent experts, it naturally aligns with exploiting sufficiently large RT with negligible overheads.

This effectively reduces WCT by using a large RT while keeping TC fixed. To further reduce TC, we allocate different resources (i.e., iterations) to each expert to take advantage of their varying convergence speeds. This accelerates overall convergence. We interpret that fast convergence can be achieved by minimizing negative interactions across different time intervals. Consequently, MDM can reduce both WCT and TC by early stopping rapidly converging experts.

We thoroughly investigate the advantages of the multi-expert approach by analyzing training patterns of diffusion models along with different time intervals (Sec. 4). In our experiments, we apply MDM on several baseline models and demonstrate the effect of MDM in terms of efficiency (i.e., training time, Sec. 5.2) and performance (i.e., generation quality, Sec. 5.3). Overall, our method improves FID by 7.7% on average, including all datasets and baselines. Furthermore, MDM offers ×2.7 - 4.7 faster training and reduces TC by 24 - 53% to reach the best baseline score.

## 2 RELATED WORKS

**Denoising diffusion probabilistic model.** Diffusion models (Ho et al., 2020; Dhariwal & Nichol, 2021; Nichol & Dhariwal, 2021; Song et al., 2020) aim to generate data through a learned denoising process. Starting from a Gaussian noise $x_T$, they iteratively denoise $x_t$ to $x_{t-1}$ using a denoising autoencoder until obtaining a final image $x_0$. We discuss theoretical backgrounds in Appendix A. ADM (Dhariwal & Nichol, 2021) proposes the optimized network architecture and proves that the diffusion model can achieve higher image sample quality than state-of-the-art GANs in several benchmark datasets (Deng et al., 2009; Yu et al., 2015). For conditional image synthesis, they further improve sample quality with classifier guidance that sacrifices diversity.

Several works focus on the time steps of the diffusion model to improve sample quality. P2W (Choi et al., 2022) identifies that diffusion models learn coarse features in later time steps, rich contents at medium, and finally, remove remaining noise at early time steps. They propose a new weighting scheme for the training objective by assigning small weights to the unnecessary noise-removal stage while assigning higher weights to the others. Since the diffusion model exhibits an unstable denoising process nearly at $t = 0$ (infinite signal-to-noise ratio), both discrete and continuous time-based diffusion models (Ho et al., 2020; Song et al., 2021b;a) truncate the smallest time step (early-stopping denoising process before it reaches $t = 0$). Soft-truncation (Kim et al., 2021) claims that a small truncation hyperparameter favors negative-log-likelihood (NLL) at the sacrifice of FID and vice versa. To secure both NLL and FID, they soften the static truncation hyperparameter into a random variable so that the smallest diffusion time step is randomly chosen at every optimization step. P2W and Soft-truncation improve the image quality by regularizing the model along time steps. However, based on our observation, they train the entire time steps at once, causing a negative influence among different time steps. Unlike these methods, our work identifies and then effectively eliminates such negative influences.

**Efficient training for generative models.** Several researchers have attempted to enhance the efficiency of generative models. Pang et al. (2020) propose a finite-difference score-matching function for score-matching generative models. Anycost-GAN (Lin et al., 2021) reduces the usage of inference resources by dynamically leveraging model parameters during inference. Similarly, DDIM (Song et al., 2020) and EDM sampling (Karras et al., 2022) aim to reduce the resources required for the sampling process of diffusion models. However, these approaches only focus on reducing inference costs, not training costs. To improve the training efficiency, LDM (Rombach et al., 2022) seeks to reduce the parameter size of the model by reducing data resolution via autoencoders. Patch-Diffusion (Wang et al., 2023) proposes a data- and resource-efficient diffusion model by generating images in a patch-wise manner. Their focus is to improve model efficacy by changing from natural images to patch images in the data domain. These approaches are orthogonal to our method as they modify the domain of data distribution.

Concurrent to our work, several methods (Balaji et al., 2022; Feng et al., 2023) utilize multi-expert fine-tuning on a pre-trained text-to-image diffusion model to seamlessly reflect the text-conditional signal. Although they utilize a multi-expert strategy (Artetxe et al., 2021; Shazeer et al., 2017; Riquelme et al., 2021), their experts share the same pretrained model as initial points for fine-tuning. This approach limits training efficiency since those require a resource-intensive pretraining stage. Furthermore, they focus on conditional generation scenarios, which enhance text-and-image alignment through fine-tuning. We (1) do not deal with pre-trained models but the training efficiency of

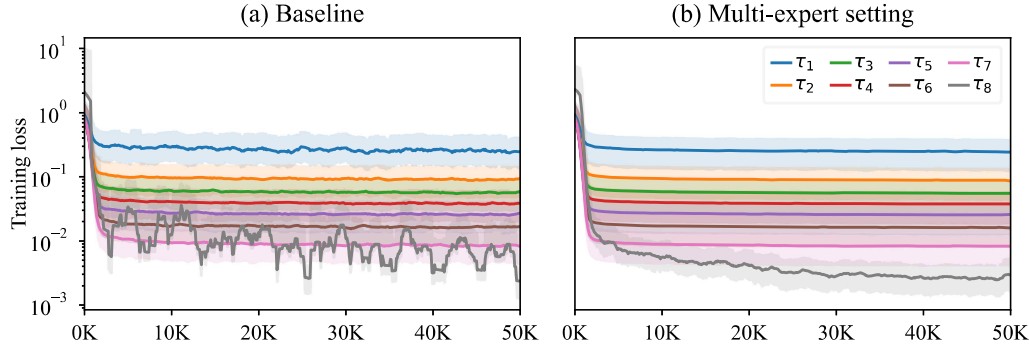

Figure 2: Training losses for (a) the baseline ADM (Dhariwal & Nichol, 2021) and (b) the eight-expert settings. To improve visualization, we average five adjacent points to filter out the noise in the graph. The color-shaded area depicts the range between minimum and maximum values for adjacent points.

the model when training from scratch and (2) target unconditional diffusion models, which affects various applications. More importantly, our method works as an add-on module to these previous researches to enhance their training efficiency.

## 3 MULTI-EXPERT DIFFUSION MODEL

We propose the Multi-expert Diffusion Model (MDM) as an efficient training solution for diffusion models. Our approach centers around two objectives: (1) partitioning the model for independent training that better aligns with a large resource throughput (RT) and (2) improving the convergence speed of each expert to reduce the total cost (TC). Our investigation (Sec. 4) reveals distinct training patterns within the diffusion model, characterized by three groups of time steps exhibiting similar training patterns. Based on this observation, we introduce a training strategy that involves three experts, each responsible for training a specific group of three-time step intervals: $\tau_A, \tau_B, \tau_C$.

Following Ho et al. (2020), we employ a denoising autoencoder to model the reverse process of the diffusion model. The learnable parameters $\theta(t)$ of MDM, given by a denoising autoencoder $f_{\theta(t)}(x_t, t)$, can be expressed as:

$$\theta(t) = \begin{cases} \theta_A, & t \in \tau_A, \\ \theta_B, & t \in \tau_B, \\ \theta_C, & t \in \tau_C. \end{cases} \tag{1}$$

$\tau_A$, $\tau_B$, and $\tau_C$ vary depending on the baseline model and the image resolution. The range of each interval determined for each experiment is specified in Sec. 5.1. The experts in MDM ($f_{\theta_A}$, $f_{\theta_B}$, and $f_{\theta_C}$) are trained independently within their designated time interval. For a fair comparison with the baseline, we initially set the maximum number of iterations $\mathbb{I}_e$ for each expert equally to $(|\tau_e|/T)\mathbb{I}_{\text{baseline}}$, $e \in \{A, B, C\}$. In this context, $|\tau_e|$ denotes the number of time steps within the interval $\tau_e$, and $\mathbb{I}_{\text{baseline}}$ indicates the total iterations for training the baseline model. Then, we assign additional iterations to the expert with a relatively slower convergence while maintaining the sum of all $\mathbb{I}_e$ equal to $\mathbb{I}_{\text{baseline}}$. Each expert's architecture remains consistently the same.

**Remarks on training efficiency.** Our multi-expert approach offers two advantages: (1) utilizing a large RT with negligible overhead and (2) faster convergence to optimal performance for each expert. These two advantages reduce WCT and TC, respectively.

Firstly, training multiple experts independently empowers us to effectively reduce WCT by employing a large RT while minimizing additional overhead. Although the baseline model can be trained on multiple GPUs (or nodes), it is limited by practical resistance, such as finite batch size (which limits the maximum number of devices used) and communication overhead between devices. In contrast, our model has three independent experts, allowing us to increase RT more effectively than training the baseline with multiple nodes, with negligible practical resistance (see overhead analysis).

Secondly, our method trains each time interval independently, thereby focusing on each distinct training pattern. This mitigates the potential negative interactions among different time steps when

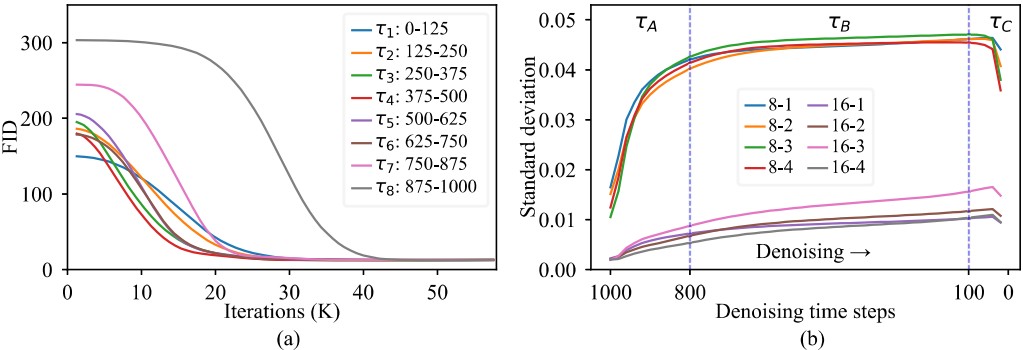

Figure 3: ADM time steps analysis. (a) Performance convergence in FID of eight experts for each time step range $\tau_i$. (b) Concentration of attention weights during the denoising process. Each legend '$r$-$o$' indicates $o$-th $r \times r$ attention layer.

Table 1: Comparison between uniform and importance sampling strategies for time step sampling in the CIFAR-10 dataset. We train the baseline (Dhariwal & Nichol, 2021) using uniform sampling and importance sampling and evaluate them after 300K iterations.

| Method | FID↓ | sFID↓ | Precision↑ | Recall↑ |
|---|---|---|---|---|
| Uniform | **12.42** | **27.34** | **0.5777** | 0.6247 |
| Importance (Nichol & Dhariwal, 2021) | 18.35 | 34.77 | 0.5532 | **0.6355** |

training the entire time step simultaneously (Fig. 2). As a result, we consistently observe that any of the three experts in MDM reach optimal parameters faster than the baseline model. Furthermore, we assign additional iterations to the experts in $\tau_A$ and $\tau_C$ due to their slower convergence compared to the expert in $\tau_B$. Our strategic allocation of training resources to the slower experts accelerates the overall convergence, reducing TC (Sec. 5.2).

**Overhead analysis.** MDM utilizes three experts, resulting in three times the number of parameters compared to the baseline. However, the model capacity remains unchanged in terms of vRAM (or other equivalent limiting devices), serving as a true bottleneck in computing resources. Training and inferring each expert is independent of each other, thus MDM does not require simultaneous vRAM access for multiple experts. The additional storage space required to store the parameters can be achieved with more affordable and sufficient options, such as flash memory. The slight increase in loading time required to transfer the model to vRAM is negligible compared to TC. Therefore, from a practical standpoint, the resource overhead associated with our method is manageable.

## 4 DIFFUSION MODEL DISSECTION

In this section, we delve into the detailed process of dissecting the time steps of diffusion models into three main groups for our MDM (Sec. 3). We analyze the training patterns of the diffusion model and conclude that the standard method of training all time steps at once hinders fast convergence (Sec. 4.1). We divide whole time steps into three groups for efficient training based on activation analysis (Sec. 4.2).

### 4.1 TRAINING DYNAMICS ANALYSIS

Each time step in the diffusion model is trained independently, and the loss scale diverges as t → 0 (Kim et al., 2021). We hypothesize that simultaneously training the entire time steps with varying loss scales (standard method) can hinder the training process. Therefore, we explore the impact of dividing the whole time steps into distinct groups and training each separately. It increases resource throughput (RT), thereby reducing actual training time (WCT). However, this alone is insufficient to reduce the total cost (TC).

To further reduce TC, we investigate the training dynamics of diffusion models across distinct time intervals. Our investigation reveals that (1) the convergence speed varies for each time interval and (2) exploiting multi-expert training across different time intervals exhibits more stable training

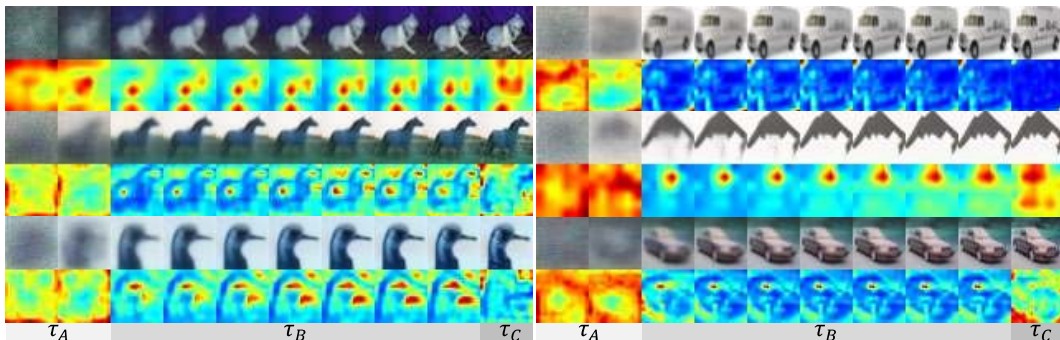

Figure 4: Visualization of the image sampling process and its attention layer weights. Odd rows depict the image prediction samples ($x_0$) obtained from DDIM (Song et al., 2020) sampling. Even rows demonstrate the attention layer's activations, normalized by dividing them with the maximum value for improved visualization.

Table 2: Comparison of training time and resource requirements. NWCT (%) and NTC (%) are normalized by 500K training iterations for ADM and P2W, and 5.0M for Soft-truncation. 'Converged' indicates the convergence point of the model. 'Best score' refers to the first WCT and TC, where the model achieves the best FID. 'Baseline equivalent' denotes the first WCT and TC, surpassing the best FID of the baseline model. For ImageNet-32, we omit 'Converged' due to significant performance fluctuations in the baseline model.

| Dataset | Method | Normalized WCT (%) | | | Normalized TC (%) | | |
|---|---|---|---|---|---|---|---|
| | | Converged | Best score | Baseline equivalent | Converged | Best score | Baseline equivalent |
| CIFAR-10 | ADM | 76.0 | 76.0 | | 76.0 | 76.0 | |
| | +MDM | 23.8 | 43.4 | **17.5** | 46.4 | 62.0 | **35.7** |
| | P2W | 44.0 | 48.0 | | 44.0 | 48.0 | |
| | +MDM | 29.4 | 39.2 | **17.9** | 48.0 | 57.4 | **36.3** |
| ImageNet-32 | Soft-trunc | - | 86.0 | | - | 86.0 | |
| | +MDM | - | 31.2 | **18.4** | - | 67.6 | **42.0** |

dynamics than the baseline. These observations motivate us to develop an efficient training technique for saving unnecessary resource usage, eventually reducing TC.

To examine the potentials of the multi-expert model, we divide the time steps into eight sub-intervals $(\tau_i)_{i=1}^8$ and assign an expert $f_{\theta_i}$. In this experiment, each expert shares the same architecture of ADM (Dhariwal & Nichol, 2021). The expert $f_{\theta_1}$ is responsible for generating the final clean image, while $f_{\theta_8}$ starts denoising from the noisy latent. We train each $f_{\theta_i}$ for $\tau_i = \{t|t \in (125(i-1), 125i]\}$ on CIFAR-10 dataset (Krizhevsky et al., 2009). We evenly assign 62.5K iterations per expert, where a total of 500K iterations are used for both MDM and the baseline. We investigate the multi-expert setting in two aspects: convergence speed and training loss.

**Convergence speed.** To measure the convergence speed of each expert, we vary the iteration for the $i$-th expert while keeping the other experts fully trained (62.5K iterations). We calculate FID between the sampled 10K images and the 10K images of the CIFAR-10 validation set. Fig. 3(a) visualizes the FID values at each iteration for each expert. Interestingly, we observe different convergence speeds for each expert. The experts of middle time intervals show the rapid convergence at around 26K iterations. In contrast, the expert $f_{\theta_8}$ converges at around 45K iterations, demonstrating the slowest convergence speed. The second slowest expert becomes $f_{\theta_1}$, which starts with a lower FID and converges at around 35K iterations.

**Training losses.** We compare the training losses of the baseline (ADM) and the multi-expert setting (Fig. 2). We discover two key findings: (1) Training losses for each time step exhibit different loss lower-bound (Kim et al., 2021), and (2) the loss of the baseline presents fluctuations, especially in $\tau_8$. Investigating each loss, the time range $\tau_1$ produces a significantly higher loss ($\times 20$) than $\tau_8$, thus largely affecting the parameter updates. However, as observed in Fig. 3(a), the time range $\tau_8$ exhibits the slowest convergence, indicating a challenging stage to train. Despite its convergence challenges, the baseline cannot focus on $\tau_8$ due to its low loss scale. In this regard, we recognize that

importance sampling proposed by (Nichol & Dhariwal, 2021) has a limited impact on performance improvement since it relies more on training time steps with higher losses without considering convergence trends. As a result, when we apply importance sampling to the baseline, we observe the performance degradation (Tab. 1).

The second observation indicates greater instability within each time interval of the baseline model compared to the multi-expert setting. Specifically, the loss for $\tau_8$ depicts significant fluctuations. This result is consistent with the previous observation that $\tau_8$ is the most challenging time interval to train. This phenomenon is significantly reduced in our multi-expert setting (Fig. 2(b)). Here, we speculate that training the entire time steps with a single model could result in sub-optimal performance due to adverse interactions among different time intervals.

### 4.2 ACTIVATION ANALYSIS FOR DISSECTION

Our analysis demonstrated that multi-expert training can alleviate the negative impacts among time steps, ultimately improving training efficiency. Now, we arrive at a question: How should we partition the intervals for developing MDM?

We focus on the attention layers within the diffusion model to derive distinct intervals of MDM. Previous studies (Caron et al., 2021; Tumanyan et al., 2022) have demonstrated that attention layers provide rich visual information, such as the semantic layout of scenes. Specifically, these attention layers selectively concentrate on structural properties among features (Caron et al., 2021). Motivated by this insight, we analyze the visual information captured by the diffusion model at each time step through attention weight analysis. For that, we leverage softmax weights within the attention layer:

$$Attention(Q, K, V) = softmax(QK^\top/\sqrt{d})V. \tag{2}$$

For each attention layer, we compute the average standard deviation of softmax weights for each image as follows.

$$\mathbb{E}_t\left[\sqrt{\mathrm{VAR}_s(softmax_s(Q_{ct}K_{cs}/\sqrt{d_k}))}\right], \tag{3}$$

where subscripts follow Einstein's summation convention. A low standard deviation implies that the weight distribution is close to the uniform distribution (e.g., 0 if all values are $1/HW$, where $H$ and $W$ are the height and width of the attention map). Conversely, a high standard deviation represents weight concentration in a specific region (e.g., $\infty$ if the distribution follows the Dirac delta function). Fig. 4 demonstrates the attention layer's activations at each DDIM (Song et al., 2020) sampling time step. Fig. 3(b) illustrates the average standard deviation of the attention layer's weights from 1K samples at resolutions of $8 \times 8$ and $16 \times 16$.

Herein, we identify two distinct transitions in terms of attention concentration. As depicted in Fig. 3(b), the first group $\tau_A$ consistently increases attention concentration. In this stage, the model generates the overall outline of the resulting image, as also reported in (Choi et al., 2022). In contrast, the second group $\tau_B$ shows minimal changes in attention concentration. The outline from the previous stage remains unchanged while incorporating additional details. Lastly, the third group, $\tau_C$, shows a rapid decrease in concentration. This is because it removes an overall noise while adding natural high-frequency details (Balaji et al., 2022). These unique characteristics are used to determine three intervals of $[\tau_A, \tau_B, \tau_C]$, allowing each dedicated expert to handle distinct training patterns. Therefore, MDM assigns three experts for three distinct intervals derived in this study.

## 5 EXPERIMENTS

### 5.1 IMPLEMENTATION DETAILS

**Dataset.** We use CIFAR-10 (Krizhevsky et al., 2009) and ImageNet-32 dataset (Chrabaszcz et al., 2017) to evaluate our model. Since our multiple experts with large parameters can be vulnerable to overfitting (i.e., memorization effects are often reported in diffusion models (Carlini et al., 2023; van den Burg & Williams, 2021)), we conduct evaluations with validation sets. The validation set comprises 10K images for CIFAR-10 and 50K for ImageNet-32, respectively.

Table 3: Quantitative evaluation. All metrics report the best FID score of each model. For CIFAR-10, the best score within the same baseline (ADM or P2W) is in bold. The best score in all experiments is marked with an underline. The standard deviation of the results is denoted by underscored numbers. MDM consistently improves FID, sFID, and Recall when applied to each baseline model.

| Dataset | Method | FID↓ | sFID↓ | Precision↑ | Recall↑ |
|---------|--------|------|-------|-----------|---------|
| CIFAR-10 | ADM | 12.42 $_{(0.15)}$ | 27.34 $_{(0.15)}$ | **0.5777** $_{(0.0057)}$ | 0.6247 $_{(0.0097)}$ |
| | +MDM | **11.42** $_{(0.14)}$ | **24.86** $_{(0.15)}$ | 0.5539 $_{(0.0055)}$ | **0.6455** $_{(0.0086)}$ |
| | P2W | 11.14 $_{(0.14)}$ | 25.32 $_{(0.15)}$ | 0.5405 $_{(0.0051)}$ | 0.6263 $_{(0.0093)}$ |
| | +MDM | **10.61** $_{(0.14)}$ | **24.74** $_{(0.14)}$ | **0.5559** $_{(0.0055)}$ | **0.6569** $_{(0.0085)}$ |
| ImageNet-32 | Soft-trunc | 9.18 $_{(0.16)}$ | 4.74 $_{(0.15)}$ | **0.6018** $_{(0.0054)}$ | 0.5966 $_{(0.0083)}$ |
| | +MDM | **8.25** $_{(0.17)}$ | **4.24** $_{(0.14)}$ | 0.5879 $_{(0.0056)}$ | **0.6020** $_{(0.0094)}$ |

**Architecture.** We applied MDM on three baselines: ADM (Dhariwal & Nichol, 2021), P2W (Choi et al., 2022) and Soft-truncation (Kim et al., 2021). ADM is the representative baseline model with widely used architectures for diffusion models. P2W is a recent training strategy tailored to diffusion models. Soft-truncation represents a universal training technique for score-based models, including both discrete and continuous time-based models. We show that our method can be combined with these baselines to improve the generation quality and reduce training resources. For ADM, we employ three attention layers at resolutions of 32, 16, and 8, with three residual blocks per resolution in Unet (Ronneberger et al., 2015). The noise schedule is set as cosine. Our model has 128 channels with 32 channels per attention head and a dropout rate of 0.3. The batch size is 128, and the learning rate is 0.0001. P2W is implemented on top of ADM. We set k=1, $\gamma$=1. For sampling, we apply DDIM (Song et al., 2020) with 50 sampling steps. We set full-time step $T$ to 1000. For the soft-truncation, we follow the identical configuration for ImageNet-32 training that uses DDPM++ (Song et al., 2021a) architecture. For ADM and P2W, we set $\tau_A = \{t | t \in (0.8T, T]\}$, $\tau_B = \{t | t \in (0.1T, 0.8T]\}$, and $\tau_C = \{t | t \in (0, 0.1T]\}$. For Soft-truncation we use $\tau_A = \{t | t \in (0.6T, T]\}$, $\tau_B = \{t | t \in (0.2T, 0.6T]\}$, and $\tau_C = \{t | t \in (0, 0.2T]\}$. We observe that $\tau_A, \tau_B$, and $\tau_C$ are consistent along with model and image resolution regardless of the training dataset. Furthermore, the attention concentration of the model (Fig.3(b)) depicts similar patterns even when we train the model using only 10% of total iterations. Thus, we can obtain time step intervals without significant overheads.

**Sample quality metric.** We use four metrics to assess the quality of the generated samples. We first employ the Fréchet inception distance (FID) (Heusel et al., 2017). It provides a consistent evaluation of sample quality based on human visual assessment, outperforming the inception score (Salimans et al., 2016). FID measures the symmetric distance on the first raw and second central momentum between the two image distributions in the Inception-V3 (Szegedy et al., 2016) latent space. To capture structural relations between the data distributions more effectively than FID, we utilize sFID (Nash et al., 2021), which evaluates spatial features of Inception-V3. We also report precision and recall on the latent distribution of Inception-V3 (Kynkäänniemi et al., 2019) as FID cannot explicitly measure the distribution coverage of the generated samples.

**Computational resources.** We train our model with NVIDIA A6000 GPU. Training ADM and P2W on the CIFAR-10 dataset with a batch size of 128 for 500K iteration takes 270 hours. Soft-truncation on the ImageNet-32 dataset with a batch size of 128 for 5.0M iterations requires 462 hours.

## 5.2 TIME AND RESOURCE EFFICIENCY EVALUATION

We evaluate the practical aspects of different models by comparing their training time and resource requirements. Tab. 2 reports WCT and TC at three key points: (1) model convergence, (2) the point of achieving the best FID, and (3) surpassing the baseline model. We set the WCT and TC of the baseline model to 100%NWCT (normalized WCT) and 100%NTC (normalized TC), respectively. We consider the model to be converged when the FID difference between consecutive points is less than 0.1 for three consecutive sampling points. Simultaneously, the FID gap to the best value should be smaller than 0.3 in the CIFAR-10 dataset. ADM converges at 76.0%NWCT, while MDM on ADM converges about 3.2 times faster (23.8%NWCT). Similarly, P2W converges at 44.0%NWCT, whereas MDM on P2W converges at 29.4%NWCT, meaning 1.5 times faster.

Table 4: Ablation results on different model combinations along time steps in the CIFAR-10 dataset. Each experiment uses ADM for $\tau_B \cup \tau_C$ and only differs in the model for $\tau_A$.

| $\tau_A$ | $\tau_B \cup \tau_C$ | FID↓ | sFID↓ | Precision↑ | Recall↑ |
|---|---|---|---|---|---|
| ADM | ADM | 12.42 | 27.34 | **0.5777** | 0.6247 |
| MDM | ADM | **11.04 (-1.38)** | **25.26 (-2.08)** | 0.5604 (-0.0173) | **0.6485 (+0.0238)** |

We also identify when our MDM reaches its best FID and the best baseline FID score. Surprisingly, MDM-equipped baselines attain the best baseline score at an average of 17.9%NWCT and 38.0%NTC, being up to 4.7 times faster. Then, MDM reaches its best performance at an average of 37.9%NWCT and 62.3%NTC, still less than the baseline best score requirements. The result is visualized in Fig. 1. In conclusion, MDM effectively reduces training time and resources because of (1) higher RT with negligible computational overhead and (2) faster convergence of each expert.

### 5.3 QUALITY EVALUATION

Tab. 3 presents the quality evaluation results, reporting the minimum FID achieved by each model. We depict generated image samples in Appendix B. Applying MDM consistently improves performance across all baselines. Notably, our approach demonstrates a significant improvement in sFID and recall compared to other metrics. To identify which local expert contributes to our model to cover more diverse structures, we conduct a simple case study. As in Tab. 4, we compare the original ADM with a partially modified ADM where $f_{\theta_A}$ of MDM is exclusively applied for time interval $\tau_A$. This investigation shows that $f_{\theta_A}$ significantly improves sFID and recall compared to the baseline. This is because the time steps $\tau_A$ play a pivotal role in shaping the overall outline (Sec. 4.2), and our independent training strategy allows $f_{\theta_A}$ to generate diverse structures without negative impact from other time intervals.

Although we can manipulate precision-recall trade-off via guidance methods (Dhariwal & Nichol, 2021; Ho & Salimans, 2021) for the diffusion model, increasing recall is known to be a more challenging problem (the guidance can improve precision by sacrificing recall while the opposite is not yet available). In this view, we can conclude that MDM is capable of capturing diverse structures that lead to notable advantages in both sFID and recall.

### 5.4 COMPARISON WITH P2W AND SOFT-TRUNCATION

Both P2W and Soft-truncation aim to improve image generation quality by exploiting the roles of different time steps. However, these methods are not suitable for increasing RT without overhead, and they suffer from adverse impacts among time steps as they train the time steps all at once. By utilizing MDM, we successfully separate the training of each time step group from the others, thereby increasing training speed and eliminating the negative impact among different groups. Our method can be applied orthogonally to both methods, which not only boosts the training speed but also brings the performance closer to the optimal bound.

## 6 CONCLUSION

This paper introduces a multi-expert diffusion model (MDM) as an efficient approach for training diffusion models. MDM capitalizes on the time-independent training nature of the diffusion model. Specifically, we carefully select three-time intervals according to activation analysis and assign a dedicated expert to each interval. Three experts of our model are trained independently on their respective time step groups. This approach allows us to increase resource throughput while minimizing the computational overhead, which effectively reduces the wall-clock time required for training full iterations. Furthermore, our multi-expert strategy enables each expert to focus solely on each designated time step without any negative impacts from other time ranges. This improves overall convergence speed and leads to a significant reduction in the total cost of training the diffusion model. As a result, our model reduces total costs by 24 - 53% and training time by 63 - 79% compared to the baselines, all while achieving the improvement in the average FID by 7.7% over all datasets.

## ETHICS STATEMENT

Improving the training efficiency of diffusion models has the potential to increase the risk of abusing diffusion models in fraud and forgery. While we successfully reduced the total cost of training diffusion models, the inference stage of diffusion models still requires significant energy consumption and computational resources.

## REPRODUCIBILITY STATEMENT

Our method works as an add-on format to the existing baselines. The baseline codes are publicly available online. Basically, we edited the code to enable multi-expert training by time step modification based on the code provided by the authors of each model. When adding our method, we made effort to maintain the principle of the code, such as model structure and hyper-parameters. When the authors provide the exact training configuration, we follow it to precisely reproduce the baseline models. The configuration for training our model is stated in Sec. 5.1.

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
