## A  BACKGROUND

### A.1  DENOISING DIFFUSION PROBABILISTIC MODEL (DDPM)

DDPM is a latent-variable generative model that gradually transforms a noise distribution into a data distribution $x_0 \sim q(x_0)$ (Ho et al., 2020). DDPM consists of a forward process $q$ that iteratively adds a noise on the data distribution, and a reverse process $p$ that iteratively denoises a noise distribution toward a final data distribution. The forward process adds a Gaussian noise to $x_t$ using a Markov process according to a variance schedule $\{\beta_t\}_{t=1}^T$:

$$q(x_{1:T}|x_0) := \prod_{t=1}^T q(x_t|x_{t-1}), \qquad q(x_t|x_{t-1}) := \mathcal{N}(x_t; \sqrt{1-\beta_t}x_{t-1}, \beta_t I) \tag{4}$$

Ho et al. state that it is possible to sample $x_t$ from $x_0$ directly, using the notation $\alpha_t := 1 - \beta_t$ and $\bar{\alpha}_t := \prod_{s=0}^t \alpha_s$:

$$q(x_t|x_0) = \mathcal{N}(x_t; \sqrt{\bar{\alpha}_t}x_0, (1-\bar{\alpha}_t)\mathbf{I}) \tag{5}$$
$$= \sqrt{\bar{\alpha}_t}x_0 + \epsilon\sqrt{1-\bar{\alpha}_t}, \ \epsilon \sim \mathcal{N}(0, \mathbf{I}) \tag{6}$$

Using Bayes theorem, posterior $q(x_{t-1}|x_t, x_0)$ is also a Gaussian distribution with mean $\tilde{\mu}_t(x_t, x_0)$ and variance $\tilde{\beta}_t$:

$$q(x_{t-1}|x_t, x_0) = \mathcal{N}(x_{t-1}; \tilde{\mu}_t(x_t, x_0), \tilde{\beta}_t I), \tag{7}$$
$$\text{where} \quad \tilde{\mu}_t(x_t, x_0) := \frac{\sqrt{\bar{\alpha}_{t-1}}\beta_t}{1-\bar{\alpha}_t}x_0 + \frac{\sqrt{\alpha_t}(1-\bar{\alpha}_{t-1})}{1-\bar{\alpha}_t}x_t \quad \text{and} \quad \tilde{\beta}_t := \frac{1-\bar{\alpha}_{t-1}}{1-\bar{\alpha}_t}\beta_t \tag{8}$$

With sufficiently large T and a well defined $\beta_t$, the latent $x_T$ becomes nearly an isotropic Gaussian distribution. Assuming this, to sample from the data distribution $q(x_0)$, we can first sample from an isotropic Gaussian distribution and then iteratively apply $q(x_{t-1}|x_t)$ to obtain $x_0$. However, $q(x_{t-1}|x_t)$ depends on the entire data distribution so it is hard to exactly compute when the data distribution is unknown. As a result, we train a neural network to predict a mean $\mu_\theta$ and a diagonal covariance matrix $\Sigma_\theta$:

$$p_\theta(x_{0:T}) := p(x_T)\prod_{t=1}^T p_\theta(x_{t-1}|x_t), \qquad p_\theta(x_{t-1}|x_t) := \mathcal{N}(x_{t-1}; \mu_\theta(x_t, t), \Sigma_\theta(x_t, t)) \tag{9}$$

The network is trained by optimizing the usual variational bound on negative log likelihood, $L_{vlb}$:

$$L_{\text{vlb}} := L_0 + L_1 + ... + L_{T-1} + L_T \tag{10}$$
$$L_0 := -\log p_\theta(x_0|x_1) \tag{11}$$
$$L_{t-1} := D_{KL}(q(x_{t-1}|x_t, x_0) \parallel p_\theta(x_{t-1}|x_t)) \tag{12}$$
$$L_T := D_{KL}(q(x_T|x_0) \parallel p(x_T)) \tag{13}$$

Ho et al. identify that training the model to predict $\epsilon$ in Eq. 6 improves sample quality than directly predicting $\mu_\theta(x_t, t)$. Therefore, $L_{vlb}$ is simplified to:

$$L_{\text{simple}} = E_{t,x_0,\epsilon}\left[||\epsilon - \epsilon_\theta(x_t, t)||^2\right] \tag{14}$$

When the training is done, we can sample from the data distribution by inserting the predicted $\epsilon_\theta(x_t, t)$ to the equation:

$$\boldsymbol{x}_{t-1} = \frac{1}{\sqrt{1-\beta_t}}\left(\boldsymbol{x}_t - \frac{\beta_t}{\sqrt{1-\alpha_t}}\epsilon_\theta(\boldsymbol{x}_t)\right) + \sigma_t\boldsymbol{z}_t, \tag{15}$$

where $\boldsymbol{z}_t \sim \mathcal{N}(0, \mathbf{I})$ and $\sigma_t^2$ is a variance which is set to $\sigma_t^2 = \beta_t$.

DDPM shows a powerful performance on image generation but is has a severe drawback of significantly slow sampling speed. To sample one image, it should feedforward a neural network for each denoising step, total $T$ times. DDIM (Song et al., 2020) accelerates the sampling speed of DDPM (Appendix A.2).

### A.2 Denoising Diffusion Implicit Model (DDIM)

DDIM generalizes DDPM as a class of non-Markovian diffusion processes (Song et al., 2020):

$$q_\sigma(\boldsymbol{x}_{t-1}|\boldsymbol{x}_t, \boldsymbol{x}_0) = \mathcal{N}(\sqrt{\alpha_{t-1}}\boldsymbol{x}_0 + \sqrt{1 - \alpha_{t-1} - \sigma_t^2} \cdot \frac{\boldsymbol{x}_t - \sqrt{\alpha_t}\boldsymbol{x}_0}{\sqrt{1 - \alpha_t}}, \sigma_t^2\boldsymbol{I}) \tag{16}$$

Consequently, the reverse process becomes

$$\boldsymbol{x}_{t-1} = \sqrt{\alpha_{t-1}} \underbrace{\left( \frac{\boldsymbol{x}_t - \sqrt{1 - \alpha_t}\epsilon_t(\boldsymbol{x}_t)}{\sqrt{\alpha_t}} \right)}_{\text{"predicted } \boldsymbol{x}_0 \text{"}} + \underbrace{\sqrt{1 - \alpha_{t-1} - \sigma_t^2} \cdot \epsilon_t(\boldsymbol{x}_t)}_{\text{"direction pointing to } \boldsymbol{x}_t \text{"}} + \underbrace{\sigma_t \boldsymbol{z}_t}_{\text{random noise}} \tag{17}$$

When $\sigma_t = \sqrt{(1 - \alpha_{t-1})/(1 - \alpha_t)}\sqrt{1 - \alpha_t/\alpha_{t-1}}$ for all $t$, the forward process becomes Markovian which means that the reverse process becomes a DDPM. When $\sigma_t = 0$, the forward process becomes deterministic and produces high quality samples much faster.

# B  QUALITATIVE RESULTS

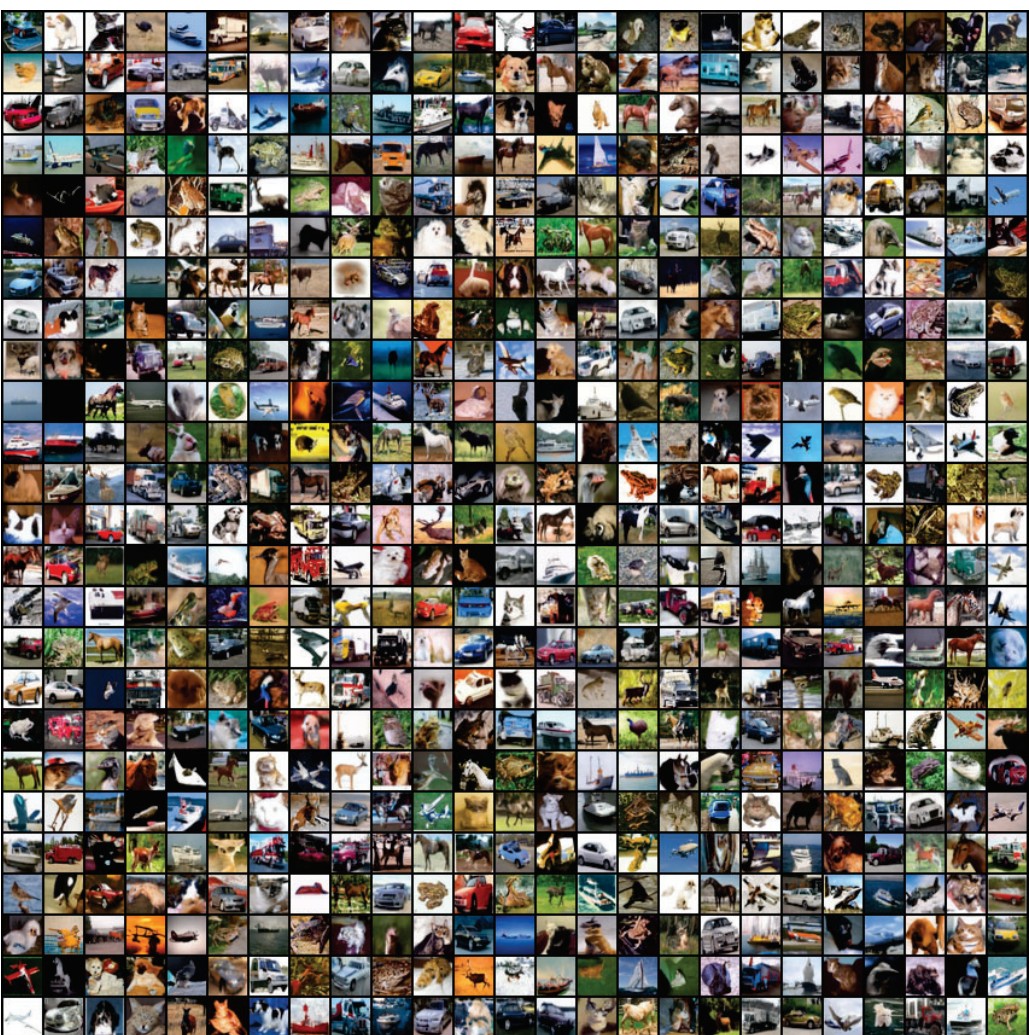

Figure A1: Image samples of MDM+ADM trained on CIFAR-10 dataset.

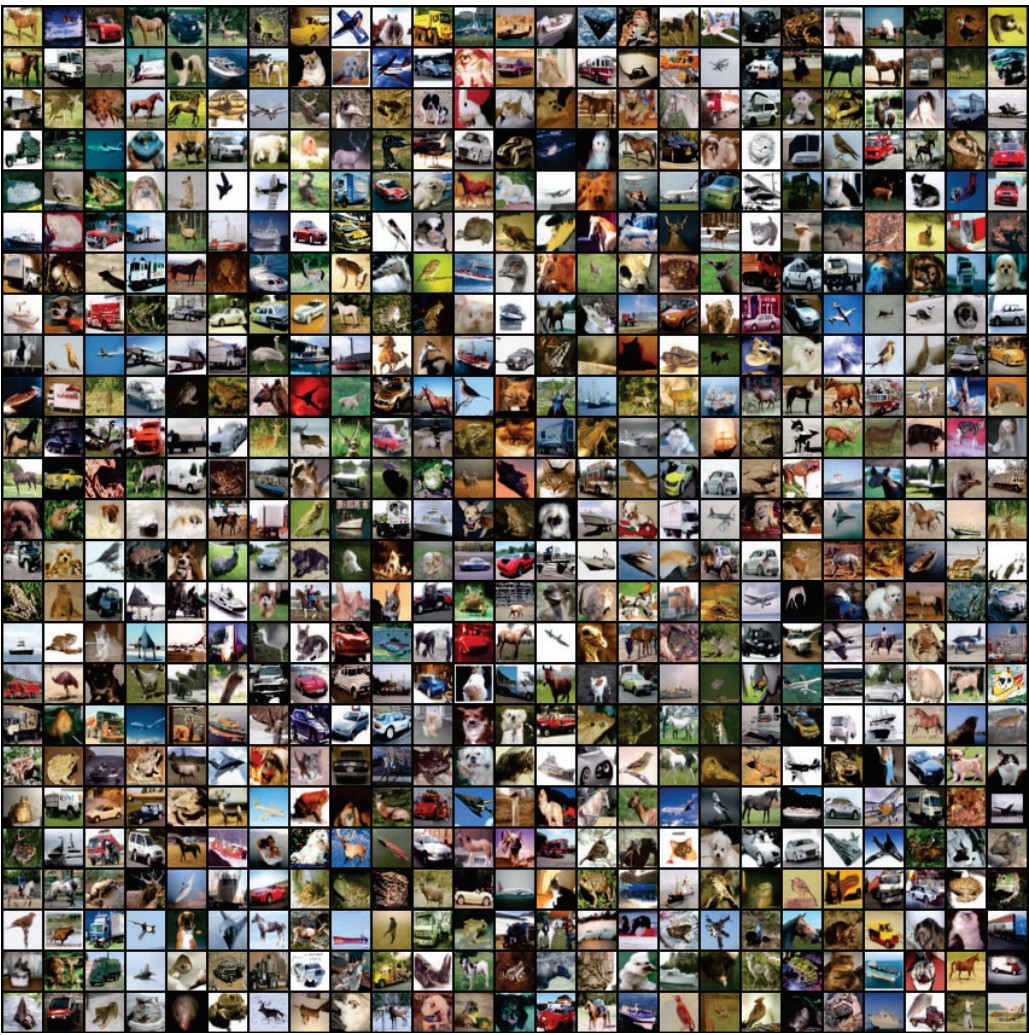

Figure A2: Image samples of MDM+P2W trained on CIFAR-10 dataset.

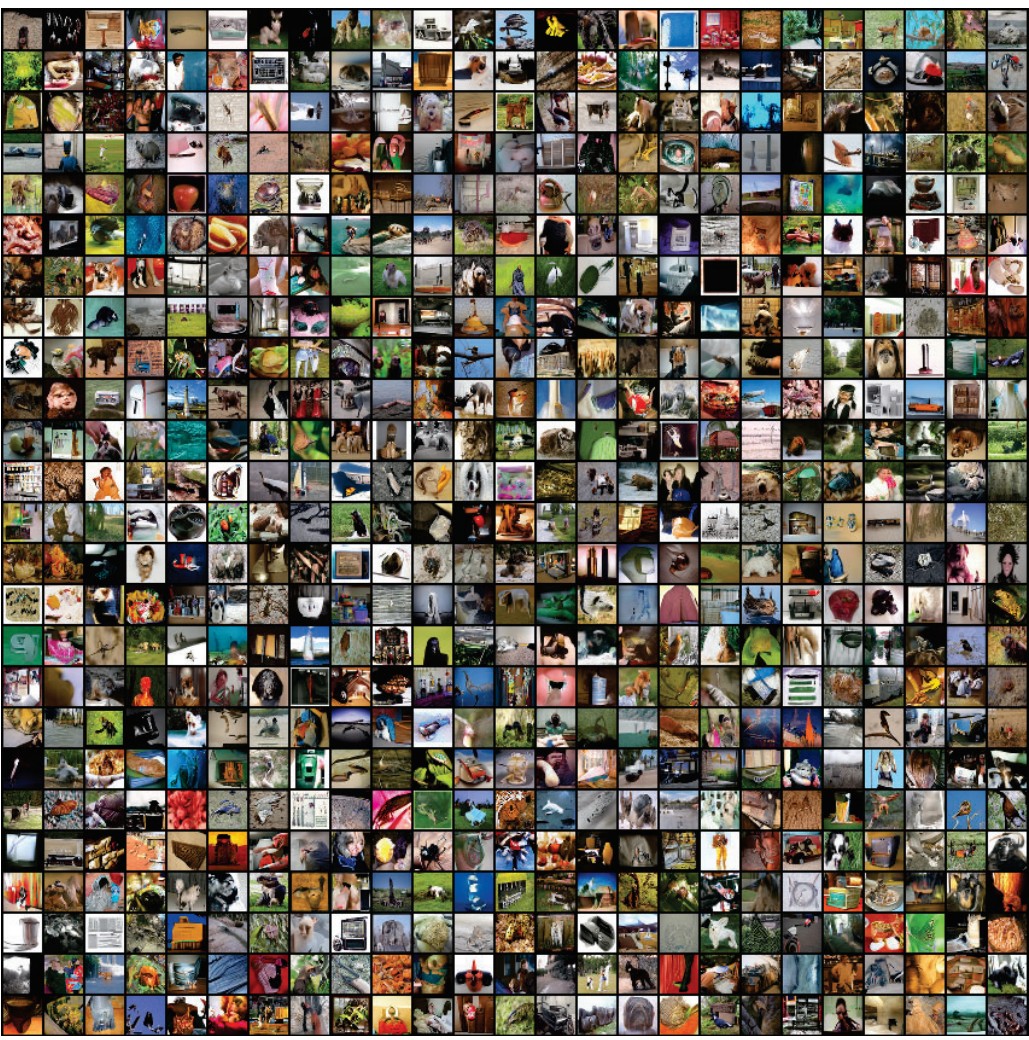

Figure A3: Image samples of MDM+Soft-truncation trained on ImageNet-32 dataset.