# OpenReview forum: "Local Expert Diffusion Models for Efficient Training in Denoising Diffusion Probabilistic Models"
_ICLR.cc/2024/Conference — ICLR 2024 Conference Withdrawn Submission_

### Official Review · Reviewer_arQF · 2023-10-16

**Soundness:** 2 fair
**Presentation:** 2 fair
**Contribution:** 2 fair
**Rating:** 3
**Confidence:** 3

**Summary:**

This paper aims to improve the training efficiency of current diffusion models. The authors propose a MDM (Multi-expert Diffusion Model), based on the observation that DMs exhibit different convergence rates and training patterns at different time steps. Experiments show improvements in the training efficiency compared to three baseline models.

**Strengths:**

- The training efficiency of the diffusion model is an important research question worth exploring.

- The proposed method shows improvements on CIFAR-10 and ImageNet-32 over baselines in terms of training time and generative results.

**Weaknesses:**

I have several concerns/questions about the current manuscript, which makes me believe that it is not yet ready for publication.

- In the introduction, the authors claim that “training each time step $x_t$ is conducted independently (Song et al., 2020)”. This statement looks very confusing and unclear to me. DDPMs are still trained in a sequential manner, my sense is that the authors may want to refer to the fact that the starting time step $t$ is essentially uniformed sampled from [1,T] (corresponding to line 3 of DDPM training algorithm)?  However, this does not mean that each diffusion time step is independent in training, which also contradicts the nature of a Markovian process.

- Following my previous point, if each interval of diffusion steps is trained in an independent manner, then how can we perform inference, especially when the denoising steps reach the intersection of those independent trained intervals? Essentially, we need to condition on the previous $x_t$ to estimate the $x_{t-1}$, but I am confused how this could be done if $t$ happens to be the end step of the proposed MDm method.

- Still in the intro, “We interpret that fast convergence can be achieved by minimizing negative interactions across different time intervals?” What are those negative interactions? These are not explained in the manuscript.

- In the captions of Fig.2, “we average five adjacent points to filter …” what are those five points? Also, the eight $\tau$ notations come with no explanations, making the figure difficult to read.

- In Table 1, why precisions and recalls are used in image generations?

- In the training dynamics analysis, are there any particular reasons for which the process is divided into 8 intervals, and the methodological design adopts three-interval designs?

- In terms of experiments, the current experiments are only performed with CIFAR-10 and ImageNet-32, which are both low in image resolutions, making the experimental results less convincing.

- Also, can authors provide insights on why the proposed MDM can improve the generative quality?

**Questions:**

Please see the Weaknesses for details.

---

### Official Review · Reviewer_qZGE · 2023-10-29

**Soundness:** 3 good
**Presentation:** 2 fair
**Contribution:** 3 good
**Rating:** 6
**Confidence:** 1

**Summary:**

This paper proposes a Multi-expert Denoising Diffusion Model (MDM) to improve training efficiency for diffusion model. The key idea is to partition the diffusion process into separate time intervals and train an expert model independently on each interval. This allows exploiting more parallel resources and faster convergence of experts on their specialized time steps.

**Strengths:**

- The paper targets a highly relevant and challenging practical problem, the training efficiency of diffusion models.
- The multi-expert approach is well-motivated through detailed analysis of diffusion model training dynamics across time steps.
- The method demonstrably provides substantial gains in training time and compute requirements.

**Weaknesses:**

- More analysis could be provided on how sensitive performance is to the time step divisions between experts.
- Only evaluated on CIFAR and downsampled ImageNet datasets. Scaling up to higher resolutions not shown.
- While the paper is generally well-organized, the layout could be refined to enhance reader engagement, particularly in terms of the placement of figures and tables relative to the corresponding text.

**Questions:**

n/a

---

### Official Review · Reviewer_8QH1 · 2023-11-01

**Soundness:** 2 fair
**Presentation:** 3 good
**Contribution:** 1 poor
**Rating:** 3
**Confidence:** 5

**Summary:**

This paper proposes to divide the time interval of diffusion models into three parts and train an expert model within each sub-interval separately. The paper carried out analysis of convergence speed and attention map statistics to determine the best way of splitting the time interval. Experimental results indicate that the MoE approach converges faster than the baseline in terms of both TC and WTC.

**Strengths:**

The paper is clearly written and easy to follow. Before getting to the designed division of time interval, detailed analysis of training dynamics and attention map statistics have been carried out, making the motivation clearer.

**Weaknesses:**

- My main concern is that this mix of expert (MoE) idea for diffusion models has already been proposed by many existing works, such as eDiff-I, RAPHAEL and SDXL, with similar conclusions and observations to the ones in this paper. The novelty seems very marginal.

- The result in Table 1 seems a bit weird. As been mentioned in "variational diffusion model", noise schedule shouldn't affect the training objective and therefore shouldn't affect the final sample quality, except for the endpoints of the schedule. Maybe that's an indication that runs in table 1 are not full converged yet.

- At the first glance of figure 3(a) it seems that t8 is challenging with very slow convergence. However, it doesn't necessarily mean that t8 is difficult, perhaps it is just that the training objective is not weighted properly across difference time steps.

- The paper mentioned that the first and third time intervals require more training iterations to converge. Have you investigated that just make these two intervals narrower such that all three intervals can converge in about the same pace? A general and automatic recipe on choosing time intervals for MoE can be more interesting.

- In Section 4.2 they proposed to use std of attention softmax logits as a measurement of concentration. But why not calculating entropy, which exactly represents such characteristic?

**Questions:**

Please see comments above.

---

### Official Review · Reviewer_Rh6m · 2023-11-04

**Soundness:** 2 fair
**Presentation:** 3 good
**Contribution:** 1 poor
**Rating:** 3
**Confidence:** 2

**Summary:**

This paper introduces dedicated expert models for different diffusion model time steps. Motivated by the fact that training independent models can maximize parallel efficiency, the authors present several observations that motivate the grouping of the diffusion time steps and training them independently for both improved efficiency and performance. The empirical effectiveness is supported by two small scale experiments.

**Strengths:**

1. Although the idea of using dedicated models for different diffusion time steps is by no means new or surprising, the authors reveal some very interesting observations that can be useful for future research. For example, the authors report different convergence speeds for time steps, where it's quite counter-intuitive that t1 demonstrates slower convergence speed compared to middle steps.

2. The paper is presented in very good writing quality.

**Weaknesses:**

My main concern with the paper is the contribution.

We use a shared model for all diffusion time steps mainly for the sake of overall model size (parameters). And it is very natural that using more dedicated parameters can improve the results of the diffusion model.

As also discussed by the authors in Section 2, using expert models for different diffusion steps is not a new idea and has been used before. And I respectfully disagree with the authors' discussions regarding the methods such as eDiff-I (last paragraph on page 2).
In this paragraph, the authors discussed two main differences/advantages of the proposed method compared to the previous methods like eDiff-I, but neither is solid.

1. Model pretraining

As discussed in eDiff-I, they do pretraining mainly to reduce the training complexity. They do not have to rely on pretraining to maintain the same final results. I don't find the pretraining of the existing work indispensable. And this can hardly be considered a disadvantage.

2. Conditional image generation v.s. unconditional image generation

The authors argue that they do unconditional image generation which 'affects various applications.' However, there is no evidence that expert-model methods for conditional image generation cannot work for unconditional image generation. And methods like eDiff-I directly target the most challenging text-to-image generation while the efficacy of the proposed method is only evaluated on two 32x32 small-scale experiments. Therefore, the contribution of this paper is further weakened.

Empirical evaluations on two 32x32 image generation tasks are not sufficient to support the idea. At least some higher-resolution unconditional experiments with 255x256 LSUN datasets are the minimum. Given the A6000 GPU used for the current results, the additional high-resolution experiments should be absolutely affordable.

**Questions:**

Please discuss the clear advantages of the proposed method compared to the existing diffusion model with expert models.